# Digital Engagement in a Contemporary Art Gallery: Transforming Audiences

**Clare Harding [1],\*, Susan Liggett [2]**  **and Mark Lochrie [3]**

1   North Wales School of Art & Design, Wrexham Glyndŵr University, Wrexham LL11 2AW, UK
2   Media Art and Design Research Centre, Wrexham Glyndŵr University, Wrexham LL11 2AW, UK
3   Media Innovation Studio, University of Central Lancashire, Lancashire PR1 2HE, UK
\*   Correspondence: clare.harding@glyndwr.ac.uk

**Abstract:** This paper examines a curatorial approach to digital art that acknowledges the symbiotic relationship between the digital and other more traditional art practices. It considers some of the issues that arise when digital content is delivered within a public gallery and how specialist knowledge, audience expectations and funding impact on current practices. From the perspective of the Digital Curator at MOSTYN, a contemporary gallery and visual arts centre in Llandudno, North Wales, it outlines the practical challenges and approaches taken to define what audiences want from a public art gallery. Human-centred design processes and activity systems analysis were adopted by MOSTYN with a community of practice—the gallery visitors—to explore the challenges of integrating digital technologies effectively within their curatorial programme and keep up with the pace of change needed today. MOSTYN's aim is to consider digital holistically within their exhibition programme and within the cannon of 21st century contemporary art practice. Digital curation is at the heart of their model of engagement that offers new and existing audience insights into the significance of digital art within contemporary art practice.

**Keywords:** contemporary art; digital shift; challenges; interdisciplinary; engagement; digital research; curators; human-centred design

## 1. Introduction

It is hard to believe that at the beginning of the 1960s it was still possible to consider artworks as falling into the category of either painting or sculpture. The rapid changes in material practices in the subsequent years saw the introduction of conceptualism, land art, performance, installation and film. The philosophical position of a 'post-medium condition' was summarised by Rosalind Krauss in her analysis of the elimination of materiality in favour of works that explored the process of embodiment; touch and the materialisation of thought processes (Krauss and Broodthaers 1999). As a consequence 'the meaning of an artwork did not necessarily lie within it, but as often as not arose out of the context in which it existed' (Archer and Paul 1997, p. 7).

Digital technologies have transformed art practices further through the introduction of new tools and new possibilities in the ways in which artworks can interact with audiences. The digital revolution has created a connected world and the consequences for art galleries and museums are immense. As Serota identified (Earnshaw 2017): "The future of the museum may be rooted in the buildings they occupy but it will address audiences across the world—a place where people across the world will have a conversation. Those institutions which take up this noting fastest and furthest will be the ones which have the authority in the future."

The key challenge of curating in the digital age is that having transformed the possibilities of art practice, galleries and curators now need to transform audiences (Chan 2016), which in turn provides its own challenges, namely

- "The Digital Shift"—The digital need as identified by arts bodies versus the lack of funding and the radical upskilling it will require presents challenges to arts organisations.
- "Digital Challenges"—How to define and explain the relationship between digital and traditional art practices and the context within which work is made to audiences of varying digital literacy.
- "Identifying Digital Potential"—How curators can take advantage of new ways of engaging with audiences, and considering MOSTYN as a case study of how a human-centred design (HCD) and an activity systems analysis approach has transformed audiences.

The research report *Digital Culture 2014: How arts and cultural organisations in England use technology* highlights the need for arts organisations to prioritise the digital in their business models. (Arts Council England 2014). Furthermore, Golant Media Ventures (Golant Media Ventures 2018) identifies that "audiences are bringing new expectations in terms of ways to connect with arts organisations and the content they produce. Without being able to adopt new digital technologies in transformational ways, arts organisations will be left behind and lose their relevance to society" (Golant Media Ventures 2018, p. 6).

MOSTYN's Director Alfredo Cramerotti addressed the key challenges identified by Chan (2016) by establishing a collaborative research project between MOSTYN, Wrexham Glyndŵr University (WGU) in September 2017, and with University of Central Lancashire (UCLan) from September 2018, both ongoing. This research investigates and identifies the existing digital content and designs a methodology for the construction of new platforms from which to develop and curate future digital outputs. The research builds upon MOSTYN's existing digital footprint and the evidence base available from its programme of activities in order to reshape perception and understanding. The ambition is for digital technology to transcend conventional structures of programming, display and engagement and to reassess approaches to the digital (e.g., as an 'add on') making it a key constituent of the core of the institution's operations and programmes.

Clare Harding joined the MOSTYN team in September 2017 as Curator of Digital Content and Media. Harding is herself a visual artist, whose practice includes incorporating digital intervention within traditional media such as oil painting and printmaking. Initially she interviewed staff and stakeholders, experimented with basic digital media and monitored social media to map out the existing organisation relationships within MOSTYN to analyse where digital tools could be of the greatest benefit. Once identified, she sought funding for projects to research and develop products to enhance audience engagement at MOSTYN. This mapping process also identified a number of tensions and contradictions within processes and systems when making a digital shift within MOSTYN. This paper will focus on the practical considerations of making such a shift within cultural institutions.

## 2. Discussion

### 2.1. The Digital Shift

MOSTYN It is one of the 67 Arts Council Wales (ACW) Arts Portfolio Wales organisations[1]. It has reputation as the foremost contemporary gallery and visual arts centre in Wales, and incorporates six gallery spaces, a studio space for engagement activities, a meeting room, a retail space and a cafe. It has two full-time staff and the equivalent of twelve part-time staff, plus a small number of volunteers. All of its operational costs are met by ACW with exhibitions and further activities funded via other grants and charitable funds, commercial activity and visitor donations. Entry to the gallery

---

[1] http://www.arts.wales/arts-in-wales/arts-portfolio-wales.

is free and MOSTYN states that it has approximately 80,000 visitors a year, most of who are visitors to the town as Llandudno only has a population of 20,000. The gallery hosts three major 'flagship' seasons of international contemporary art every year. MOSTYN also runs a series of engagement events throughout the year. This can involve working with schools, families, public talks, 'gifted and talented' teenager programmes and specialist art courses. MOSTYN activities are promoted through their website[2] and via social media. In addition, many aspects of the gallery programme are promoted via local press and periodicals, specialist art press and, occasionally, regional television coverage.

MOSTYN understood that the introduction of digital technologies in public art galleries has the potential to result in tangible benefits in terms of audience engagement and financial performance. However they also recognised that in common with many other cultural institutions, they had significant practical considerations to address. According to research, 70% of arts and cultural organisations cite a lack of funding and time, and over a third feel that they do not have the in-house skills, IT systems or the necessary expert advice to meet their digital aspirations (Digital R&D Fund for the Arts 2014).

As Chan identified, "digital transformation is really about something else that often isn't openly talked about—transforming audiences. Sure, we might change work practices along the way, but really digital transformation efforts are really in the service of visitors wherever they might be." (Chan 2016). Therefore at the heart of the transformation process, and Harding's PhD research, was one very simple question: what do visitors want?

Considering this question led to Harding identifying the problem of there being no common definition of engagement at MOSTYN. This led to further questions; is it essential people look at art to engage? Do people engage when they come in but only to buy a gift card? Is visiting MOSTYN's website the same as visiting their physical gallery in terms of the value of engagement? Without defining engagement, both in MOSTYN's organisational terms, and from the perspective of audiences, how can MOSTYN define its purpose? More importantly perhaps, how do MOSTYN's audiences define engagement? What function do they expect MOSTYN to provide for them, particularly in a digital age when a wealth of culture is available 24/7 via just a few taps on a screen.

However, even if the organisation was completely clear on its definition of engagement, and that of its audiences, there are currently no validated tools to measure such engagement. This is a gap which Ross (2014) also acknowledges: "There is a lack of consistent practice and standards in the museum computing field in respect of user profiling, motivation, participation and behaviour metrics" (Dawson et al. 2004; Haley Goldman and Haley Goldman 2005).

Research by Golant Media Ventures (Golant Media Ventures 2018) suggests that it is an issue that many non-ticketed cultural venues struggle with. At MOSTYN audience data is limited to a simple footfall door counter and anecdotal information provided by the engagement team. Online analytics only provide data on those who engage with a website; they are unable to provide demographic information for anyone who does not visit or who opts to keep their personal information private. This is a significant issue for organisations: without valid data they cannot measure the impact of their activities, compare their activities to other organisations, reflect and iterate processes nor build on successes. Harding identified that it was also a significant opportunity for digital transformation at MOSTYN.

### 2.2. The Digital Challenges

Digital media can be manifested in three possible ways: production, exhibition, and distribution.

MOSTYN demonstrated the symbiotic relationship between digital and other more traditional practices in their exhibitions of the work of Josephine Meckseper and Louisa Gagliardi in November 2018—March 2019.

---

[2]  www.mostyn.org.

Meckseper's work (e.g., Figure 1) combines paintings and found objects with her own images and film footage of political protest movements. Her installations and vitrines 'by simultaneously exposing and encasing common signifiers, such as advertisements and everyday objects, next to abstract paintings and sculptures create a window into the collective unconscious of our time.' (Cramerotti 2018). She uses digital media as a tool of production, exhibition and for her films, distribution, to rearrange and interrupt the contextual settings we associate with her found objects and images.

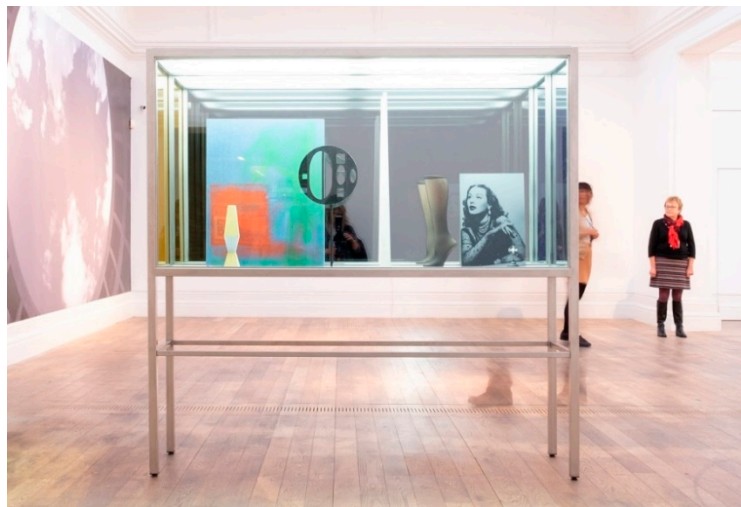

**Figure 1.** Josephine Meckseper. The Story of Mankind, 2014 (Installation). Mixed media in stainless steel and glass vitrine with fluorescent light and acrylic sheeting. Photo courtesy of Dilys Thompson.

Gagliardi's work (e.g., Figure 2) challenges the construction of an image, and of our society, in the digital age. Her work uses digital media as a tool of production, exhibition and distribution. Despite looking like traditional paintings, they are digitally printed with materials more akin to the advertising industry and set up a dialectical relationship with the history of painting. She uses a freehand digital illustration tool to 'paint' images, printing them onto PVC and then adding hand painted elements such as brush stroke marks using varnish and nail varnish to add to the simulacrum of traditional painting. 'As a whole, the appearance of her pieces is caught in a state between human and machine, reflecting the confused, surreal tone of much of the images and world's she portrays' (Carr 2018).

These artists use digital means to produce work which, in all likelihood, differs in the way the viewer understands digital in relation to their intentions. There are other examples at MOSTYN where exhibited works may be created using various media but rely on digital means of display, i.e., a looped film of a performance.

An example of this was the 2017 exhibition of Mladen Bizumic's work 'Kodak Employed 140,000 People. Instagram 13', which traced a timeline of Kodak's development from its founding in 1880 to its closure in 2012. His work 'act as a lens through which to consider much larger concepts—how the capturing of images, and the technology that enables this, influences not only aesthetic, social and economic relations, but also the resulting effects when they are replaced and taken out of the picture' (Cramerotti and Carr 2017).

Other artworks rely on technology to interact with audiences to realise the intentions of the work such as Paul Granjon's robotic art installation (Granjon 2016) that featured a robot slightly larger than your average dog that wandered around the gallery talking to itself and to the visitors who get close to it (Granjon 2016). Without an audience, there is no engagement. Ensuring audiences understand digital as a means of production, exhibition and distribution, and thus are able to fully engage with the art shown presents additional curatorial challenges.

Historically, and in common with many contemporary art galleries, MOSTYN has preferred to keep wall-based interpretation materials to a minimum. Digital media may require more contextual referencing for audiences, but MOSTYN is also considering whether an audience attracted by digital media would be happy to use more digital interpretation tools.

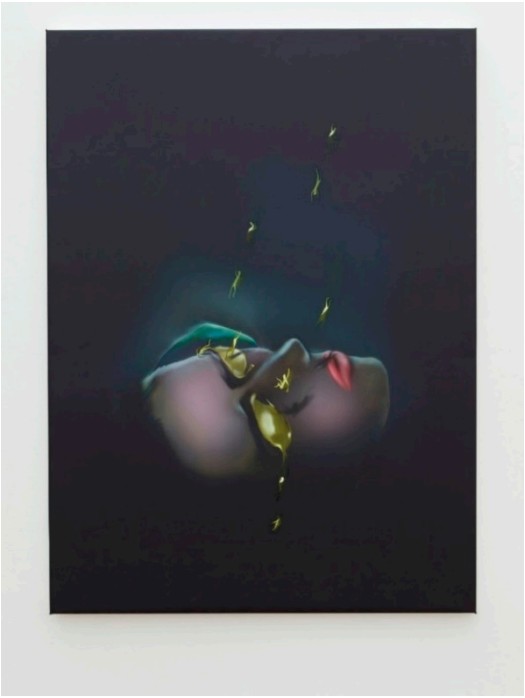

**Figure 2.** Louisa Gagliardi. Under the Weather, 2087. Gel medium, ink on PVC. Photo courtesy of Dewi Lloyd.

However digital artworks still have a hierarchy of production and accessibility just as with traditional art forms. Sixty-six percent of adults in the UK may now own a smartphone (Ofcom 2015), but that leaves 33% without them, as well as those who have older phones, that cannot access the 4G (soon to be 5G) network or do not wish to incur data charges. In addition, widespread usage of smart phones within the gallery may put pressure on the institutions' WiFi network, limiting or slowing other business activities as well as creating a sub-optimal viewing experience.

Data security scandals over recent years mean that digitally literate audiences are also increasingly sensitive about the security of their personal data, (e.g., NBC News 2019). In the UK the General Data Protection Regulations (GDPR) govern data access and sharing and galleries need to provide audiences with the reassurance they require to share their data and ensure artwork complies with legislation and ethical considerations. Branger_Briz's work 'A Charge for Privacy' was questioning where this comfort level resides for audiences back in 2011 (BrangerBriz.com 2011).

Art institutions can provide access equipment and training on how any technology involved in the display of artworks so not to deliberately exclude certain audiences from participation. However invariably there are costs associated with the provision of such equipment, as well as its security and maintenance. The challenge is how technology can be used to provide equity of experience, not imbalance.

MOSTYN are looking carefully at how technology can be used to widen participation and have been inspired by the curatorial collective Lungs Project, an activism platform that promotes the work of underexposed artists. The project builds a community of early career artists and 'adopts various publishing models as curatorial praxis to highlight more democratised and decentralised forms of producing and disseminating knowledge' (LungsProject.org 2019). Digital innovation has the capacity

to bring in missing voices to galleries, museums and collections, through the personalisation and interpretation of materials where the audience becomes the curator.

### 2.3. Challenges of Digital to Curators

Artists have always used fabricators to assist in making works and questions of authorship and attribution have been central to the discourse of art production which now has to consider artworks produced by 'digital fabricators'. Is the artwork the 3D printed object, the original design of the object within a piece of software, or the original concept by the artist? There are particular rules and systems for how to number an edition of printworks. How do you attribute, and value, an infinitely producible object that could be produced anywhere globally that has the necessary technology? This is arguably the ultimate test of the theorist Barthes' notion of The Death of the Author (Barthes 1987) and the democratisation of the reading of an artwork.

Allowing the audience to 'see behind the scenes' with digital art and understand the processes involved in its creation can be engaging for audiences as well as provide credit to the many stages and individual talents required to produce technologically complex work.

The rapid pace of technological change requires curators to keep up to date with their understanding of the technology and skills involved so they can share this knowledge to audiences.

An interesting and engaging approach was devised by The Victoria and Albert Museum, London (V&A) for their major exhibition Videogames: Design/Play/Disrupt (2018/9)[3] that explored the design and culture of contemporary video games. The exhibition investigated the work of designers, player communities and platforms and the critical conversations that surround the medium of video games. It showed how major technological advancements in the mid-2000s (i.e., broadband, social media and smart phones) changed the way videogames were designed and played, and how the language of gaming can be used to participate in social and political debates. The exhibition's curator Marie Foulston described their approach as collecting the 'debris'—sketchbooks, drawings and early storyboards—to convey the engineering, mechanics and artistry behind the games.

Recent trends in entertainment media such as Netflix, Spotify and iPlayer have enabled consumers to curate their own cultural 'playlists'. Research at MOSTYN has sought to understand how their audiences use such technology at home and whether they see value in using such technology within an art gallery. However this raises further questions about the value of the 'traditional' curator: to what degree do galleries want the public to curate their own content? Should visitors' wishes over-rule those of the curator if they are more interesting to them?

The Tate Collective is the participatory young people's programme run by Tate galleries. Evidence suggests that traditional boundaries between media is less important to younger audiences who may wish to include non-traditional media within exhibitions i.e., Whitworth Young Contemporaries: And Now We Are Plastic (January–April 2017) (The Whitworth Art Gallery 2017). MOSTYN's research continues to gather audience insight alongside reviewing such programmes and will reflect on these findings in due course.

### 2.4. Challenges of Digital to Cultural Organisations

Whilst platforms such as Facebook and Instagram may classify and measure engagement as a simple double tap to 'like' a post, what does engagement with a cultural institution look like in a digital age? Is visiting the website the same as visiting the building? Digital events instead of actual ones are more cost-effective and have the potential for global reach, but if people do not come through the door, you miss out on other perhaps critical commercial opportunities. Just as BBC Three moved online in

---

[3]　Victoria and Albert Museum 'Videogames: Design/Play/Disrupt' exhibition (8 September 2018–24 February 2019). London. Available online: https://www.vam.ac.uk/articles/about-videogames-exhibition (accessed on 20 March 2019).

2016[4], would being successful at engaging online audiences mean that funders could question the need for a 'bricks and mortar' presence?

The costs of equipment, staff training and security, the obsolescence of technology and the costs involved with staying current, all against a political backdrop of a lack of capital grant funding available for arts organisations, means that galleries have got to be both wily and innovative. Valuable lessons can therefore be learnt from the 'alternative' development communities: hackers, makerspaces and open source software.

As Chayka (2012) identifies the many positives that come with such an approach: "Not only does this philosophy allow for a stronger community of artists and creative workers, it also infuses a greater energy into the artistic discourse, a willingness to publicly experiment and a collective capacity that makes larger projects easier to undertake and accomplish." A very successful example of this approach is Furtherfield Commons (2019).

At MOSTYN Harding included makers, hackers, software, digital and public space designers, artists and creatives within her research team and as research participants and sought to ensure a wide ranging, innovative and interdisciplinary approach to the problems identified.

## 3. Identifying the Digital Potential at MOSTYN

Harding's early literature reviews indicated that existing research seems often to apply technology within a gallery setting and then measure audience reaction to it (Hartley 2015; Ross 2014; Walker 2015). Harding identified a crucial gap within both this field of knowledge, and within MOSTYN, namely to start with audiences and understand their beliefs, behaviours, preferences and insights then, if applicable, to use digital means to deal with their identified issues and fulfil the wider organisational ambitions of MOSTYN. In short, this means applying a design thinking approach.

The EDGE (Experiential Display to Generate Engagement) research project was codesigned by the research partners to gain insight into what the public want from cultural institutions and MOSTYN's purpose as a public art gallery in the face of rapidly developing, culturally competing technologies. The first stage of EDGE was a human-centred feasibility study looking at how audiences want to interact with a public art gallery in the face of rapidly developing, culturally competing technology. The project would then explore new and authentic ways in which MOSTYN could communicate with audiences.

A human-centred design approach was taken to create a new analytical framework to understand audiences and establish themes, patterns and behaviours at MOSTYN.

Applying cultural-historical activity theory (CHAT) (Vygotsky 1978), Harding created an organisational activity map (Figure 3) to understand the context and origination of MOSTYN's identified issues and their dialectical relationships. CHAT theory espouses that consciousness is essentially subjective and shaped by the history of each individual's social and cultural experience and that "the events that occur during the activity and the consequences of the activity can qualitatively change the participant, his/her object and motives for participation, the social environment of the activity and the activity itself" (Kaptelinin 2005; Rogoff 1995 cited in Yamagata-Lynch 2007). As Kaptelinin identifies, such an approach helps researchers "to ask right questions-rather than providing ready-made answers" (Kaptelinin 2005 cited in Kaptelinin 2005).

---

4   https://www.bbc.co.uk/news/entertainment-arts-35578867.

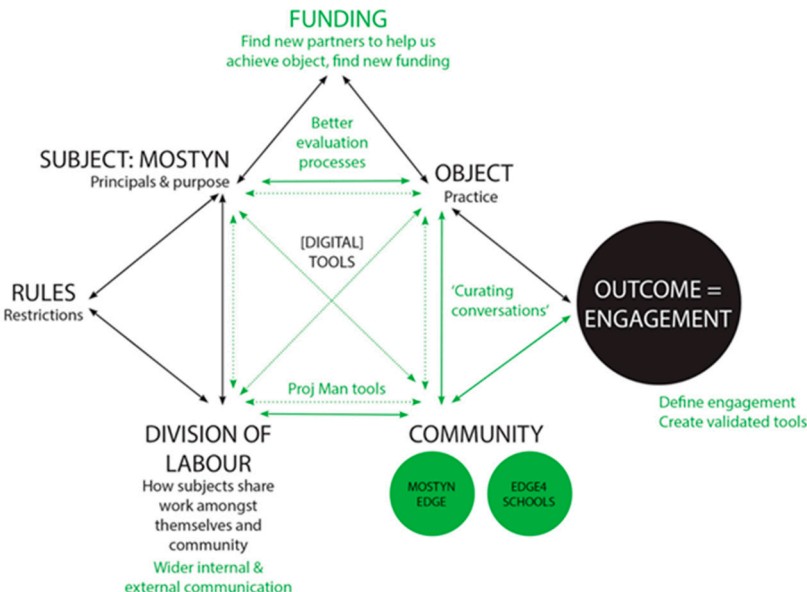

**Figure 3.** Opportunities for digital interventions at MOSTYN within proposed experimental Activity System 1 identified by Harding as part of this research, 2018.

A series of engagement events were held:

1.  A MOSTYN staff workshop role play and pilot public tasks before the research team facilitated
2.  A public workshop with 18 audience members, staff and stakeholders. Participation was voluntary following a call out via MOSTYN's visitor database and social media. The workshop to deployed and tested multiple agile, lo-fi prototypes for tools that participants felt would improve their MOSTYN visitor experience. Insights were shared about their recent visits, experiences and knowledge of MOSTYN through a series of interactive tasks. Findings were then organised and ranked by participants to establish a hierarchy of needs. Lo-fi prototypes of possible solutions were then created, using cardboard, paper and string and these were presented to the rest of the group.
3.  A public consultation was held within MOSTYN to present the lo-fi prototypes to audiences for further comments, along with a small selection of tasks taken from the public workshop to triangulate findings.
4.  A rapid, intensive problem-solving design exercise (Design Sprint) was held with MOSTYN staff to take public ideas and insight and combine them with organisational need to develop a new digital interface within MOSTYN.

Participants within these consultations were overwhelmingly positive about the researches processes used. Sometimes difficult, but always constructive, discussion took place with participants and the research team. The research design proved to be a mutual education process with participants gaining insight into how MOSTYN works, and the research team gaining authentic understanding of audience needs.

Early findings show that the public audiences who participated in these exercises are broadly happy with activities at MOSTYN and support the gallery in its digital objectives. However, they do want more; more art, more talks, more artists featured, more engagement, more interpretation materials and a more significant presence in the town. MOSTYN is acting upon these wishes through the co-creation of a series of pilot studies that incorporate digital and nondigital cost-effective solutions with the researcher's interdisciplinary team.

## 4. Conclusions

It is anticipated that the EDGE model could be used as a tool kit for other organisations utilising a human-centred design process to place audiences at the heart of the decision-making processes involved in digital strategies in public art galleries. The audience also play an important role when following this approach as they are placed at the centre, on the boundaries and everywhere in between. The audience is key to the decisions made throughout the project, which is really important when exploring ways of understanding what an audience of the future might look like.

Our research suggests that an interdisciplinary approach is crucial if art galleries like MOSTYN are to succeed in the digital age. Partnerships such as the one between MOSTYN, WGU and UCLan are essential in ensuring that digital can disrupt and break hierarchies, change power relations, supplement other structures within a gallery environment and be more than just an 'add on' to existing programmes. Collaborating organisations can share equipment, space and knowledge, to codesign and self-direct projects.

"Futurescoping" is a term process used in film making to review the script to ensure objects and words are not obsolete by the time the film eventually makes it to audiences. The same applies to public art galleries and it is important to include user testing and audience testing when staging exhibitions and be flexible enough to respond to findings.

At the heart of EDGE is an interdisciplinary team of researchers, designers and creative technologists. Through thinking differently this collaboration has yielded innovative approaches that have enabled MOSTYN to reach its audiences in novel ways. This collaborative approach has brought innovation from key disciplines to share insights and best practices to inform the process and final prototype.

**Author Contributions:** Conceptualization, C.H.; Methodology, M.L.; Software, M.L.; Formal Analysis, C.H.; Investigation, C.H., M.L.; Resources, C.H., M.L., S.L.; Data Curation, C.H., M.L.; Writing—Original Draft Preparation, C.H., S.L.; Writing—Review & Editing, C.H., S.L.; Visualization, C.H.; Supervision, S.L.; Project Administration, C.H.; Funding Acquisition, C.H., M.L. C.H., S.L. and M.L. have approved this submitted version and agree to be personally accountable for the author's own contributions and for ensuring that questions related to the accuracy or integrity of any part of the work are appropriately investigated, resolved, and documented in the literature.

**Funding:** This research was funded by the Arts Council of Wales and InnovateUK.

**Acknowledgments:** The authors would also like to acknowledge the invaluable contribution of Adrian Gradinar to the research.

**Conflicts of Interest:** The authors declare no conflict of interest.

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
