# Peer review of "Digital Engagement in a Contemporary Art Gallery: Transforming Audiences"

_arts_

Round 1
Reviewer 1 Report
This submission is timely, ambitious, significant and overall clearly presented, and the author(s) should be commended for this. However it could be significantly improved by:
1) reducing the length of cited text and more smoothly integrating them/their ideas into the article;
2) checking for sentence fragments, typographical and grammatical errors (see lines 11 and 12 of the abstract; p. 3, lines 110, 112, 121, for examples);
3) clearer signposting and use of transitions to guide the reader through the sections of the text;
4) referencing (within the text) and providing a clear explanation of Figure 3.
Author Response
All points have been addressed. Thank you for your constructive comments.
Reviewer 2 Report
55. Recommend adding date citing when the collaborative project between MOSTYN, WGU and UCLan took place.
57 - 60. Change in tense from one sentence to another. Keep it consistent i.e. 'The research built upon MOSTYN's...'
94. No need to add 'too' to the end of the sentence.
95. Check grammar/tense: instead of 'citing' and 'feeling' change to 'cite' and 'feel'.
120-121. 'Online analytics only provide data on who they are engaging with online; they do not know who is missing.' This sentence needs rewriting and expanding on as the meaning is not clear.
122. missing 'a'. without [a] valid date.
130. delete 'explores'.
140. Delete 'too' to say 'Her work uses digital media'.
142. Spelling. 'advertising industry and set up'.
151. Should say 'There' not 'The'.. 'There are other'
152. Not a great sentence - please amend. Substitute 'any media' for 'various media and which rely on digital means to...' changing present them as such' as this does not make sense to reader.
164. Change 'means' to 'presents'.
183. Change 'of course' to 'invariably'.
187. Missing word. Should read 'inspired by the'.
192. Poor sentence structure. Amend to ' interpretation of materials where the audience becomes the curator'.
206. missing word. Should read: 'understanding of the technology.'
209. 'This exhibition' should read ' The exhibition'.
212-213. Poor sentence structure - needs rewriting. Should read 'videogames were designed, played and debated, the exhibition also showed how the language of gaming encouraged audiences to participate in social and political debates.'
223. Should read 'media is less'. Also less important to who - young people or Tate Galleries - please add audience you are referring to.
228. Amend to 'may classify and measure engagement as...'
246. Amend to 'participants sought to ensure..'.
249 - 256. '3. Identifying the Digital Potential at MOSTYN'. This is a crucial point in the paper yet this section is not very clear. Suggest further detail and breaking down of the argument. Currently poor sentence structure.
297. Add a comma after 'research team,'.
299. Change 'getting to 'gaining'.
312. missing words. 'exploring ways [of[ understanding'.... 'the future [might] look like.'
324. Extra word. Remove 'that' from 'this collaboration has yielded'
326. Remove 'all' from 'key disciplines'.
Author Response

(The authors gave the same response as above.)
